# Evaluation of Meat Meal as a Replacer for Fish Meal in Diet on Growth Performance, Feed Utilization, Chemical Composition, Hematology, and Innate Immune Responses of Olive Flounder (*Paralichthys olivaceus*)

**Md. Farid Uz Zaman [1], Ran Li [1] and Sung Hwoan Cho [2],***

1   Department of Convergence Study on the Ocean Science and Technology,
    Korea Maritime and Ocean University, Busan 49112, Republic of Korea
2   Division of Marine Bioscience, Korea Maritime and Ocean University, Busan 49112, Republic of Korea
*   Correspondence: chosunh@kmou.ac.kr

**Abstract:** This study aims to evaluate the dietary replacement effect of various levels of fish meal (FM) with meat meal (MM) on the growth, feed utilization, chemical composition, hematological parameters, and innate immune responses of olive flounder. A total of 360 juvenile fish (initial weight of 14.7 g) were randomly assigned to 18 flow-through containers. The control (MM0) diet included 65% FM. Then, 10%, 20%, 30%, 40%, and 50% FM in the MM0 diet were replaced with MM, referred to as the MM10, MM20, MM30, MM40, and MM50 diets, respectively. The fish were hand-fed to satiation daily for 56 days. Weight gain, the specific growth rate, the feed efficiency ratio, and the protein efficiency ratio of fish fed the MM0 diet were statistically greater than those of fish fed the MM30, MM40, and MM50 diets, but not statistically different from those of fish fed the MM10 and MM20 diets. To incite the maximum values of weight gain and the specific growth rate (SGR) of the fish, an estimated 7.0% of FM substitution with MM in diets was required according to regression analysis. However, the feed consumption, protein retention, hematological parameters, and innate immune (superoxide dismutase and lysozyme activities) responses of the fish were not statistically impacted by the dietary replacement levels of MM for FM. In conclusion, the feed ingredient grade of MM can substitute FM by up to 20% in the diet without causing any negative impact on the growth, feed consumption, feed utilization, or innate immune responses of olive flounder.

**Keywords:** olive flounder; fish meal replacement; meat meal; regression analysis; innate immune response

## 1. Introduction

Aquaculture production is now commonly used as a significant source of animal protein for human consumption. With an annual growth rate of about 6%, aquaculture has emerged as the most rapidly growing agricultural sector in recent decades [1]. According to statistics, world aquaculture production reached 87.5 million metric tons in 2020 and is anticipated to exceed 106 million metric tons by 2030 [2]. Olive flounder (*Paralichthys olivaceus*, Temminck and Schlegel, 1846), popularly known as bastard halibut, is one of the main commercial aquaculture fish species in the Eastern Asian countries, such as the Republic of Korea (*hereafter*, Korea) and Japan [3]. In 2021, olive flounder aquaculture production in Korea was 41,776 metric tons, accounting for 46.7% of the total aquaculture production in that year [4].

With the expansion of global fish aquaculture, the demand for fish feeds is increasing dramatically. The most expensive nutrient in fish feeds is protein, with inclusion levels typically ranging from 30 to 50% [5]. Fish meal (FM) has been employed as the primary protein source in aquafeeds due to its high protein and well-balanced amino acid (AA)

content [6]. Nevertheless, the decreased supply has resulted in a decline in FM production, which has led to prices of ca. USD 1500 per metric ton [7]. The high price of FM as well as the adverse environmental effects of fish farming sites may limit the anticipated expansion of aquaculture.

Some studies [5,8] have suggested minimizing the addition of FM to fish feeds based on concerns regarding economic and environmental sustainability. Therefore, it is necessary to identify and look for a replacement that is inexpensive, readily available, and has well-balanced AA profiles for FM in fish feeds [9,10]. Recently, there has been an increasing number of studies on a novel substitute for FM in fish feeds and research will continue to expand worldwide. There has been growing interest in using various animal and plant sources in fish feeds to alleviate the pressure from FM usage [9,11–13]. Nevertheless, when utilized as sources of protein for fish diets, most plant protein sources have several issues, including imbalanced AA, declined palatability, as well as a variety of antinutritional factors [5,6,14]. These problems could also adversely impact feed intake, digestion, nutrient absorption, and the health conditions of fish [6,14]. Therefore, a candidate must have certain characteristics to be a feasible alternative source of FM in fish diets, such as being rich in protein, having a low price, worldwide availability, and no adverse impact on fish health.

Meat meal (MM) is a by-product meal generated by slaughterhouses or meat processing plants from land animals. It is not only rich in protein but also reasonable in price, and has been widely used in the production of various fish diets during the last few decades [9,15–20]. Therefore, it is predicted that feed costs could be lowered by replacing FM with MM in the diets of olive flounder and other carnivorous fish species. Reference [18] indicated that 60% of white FM could be successfully substituted with pet-grade MM (80% crude protein and 12% crude lipid) with supplementation of the essential AAs of lysine, methionine, and tryptophan, which are likely to lack or be deficient in MM in an olive flounder diet at a mean rearing temperature of 20 °C. Reference [15] reported that a 40% replacement of FM with feed ingredient-grade MM in a juvenile olive flounder diet could be implemented at a mean temperature of 12.3 °C during the winter season. Our previous study [16] also found that FM up to 40% in a diet without the supplementation of AAs could be successfully replaced with pet-grade MM (80.3% crude protein and 13.9% crude lipid) without deteriorating the growth of olive flounder. Therefore, the suitability of MM as a replacement for FM in fish feeds appears to vary depending on the fish species, MM quality, supplementation of AAs, and other experimental conditions, including the water temperature. In addition, the substitutability of FM with MM in olive flounder feeds is still controversial.

The current study, therefore, aims to evaluate the effect of replacing various levels of FM with MM in olive flounder feed on the growth, feed consumption, feed utilization, hematology, chemical composition, and innate immune responses of olive flounder.

## 2. Materials and Methods

### 2.1. Diet Formulation

Feed ingredient-grade MM (65.0% crude protein and 10.9% crude lipid) was purchased from Daekyung Oil & Transportation Co. Ltd. (Busan Metropolitan City, Korea). Six experimental diets were formulated based on the nutritional requirements of olive flounder (Table 1) [21,22]. Sixty-five percent FM (anchovy meal/sardine meal = 1:1) and 12% dehulled soybean meal were included as the primary sources of protein in the control (MM0) diet. The MM0 diet also contained 16.5% wheat flour as the carbohydrate source, and 2% fish oil and soybean oil each as the lipid sources. The graded levels (10%, 20%, 30%, 40%, and 50%) of FM were replaced by MM, referred to as MM10, MM20, MM30, MM40, and MM50, respectively. All experimental diets were formulated to be isonitrogenous at 55.0% and isolipidic at 10.5%. All ingredients of the diets were thoroughly blended with water at a ratio of 3:1. After that, the mixed ingredients were pressure-pelleted with a laboratory pellet extruder and dried at 40 °C in an electronic drying machine (SI-2400,

SIN IL Drying Machine Co. Ltd., Daegu Metropolitan City, Korea) for 24 h. Finally, all experimental pellets were kept in a refrigerator at $-20\ °C$ until use.

**Table 1.** Ingredients of the experimental diets (%, DM basis).

| | **Experimental Diets** | | | | | |
| | **MM0** | **MM10** | **MM20** | **MM30** | **MM40** | **MM50** |
|---|---|---|---|---|---|---|
| Ingredient (%, DM) | | | | | | |
| Fish meal (FM) [1] | 65.0 | 58.5 | 52.0 | 45.5 | 39.0 | 32.5 |
| Meat meal (MM) [2] | | 7.4 | 14.7 | 22.0 | 29.4 | 36.7 |
| Dehulled soybean meal | 12.0 | 12.0 | 12.0 | 12.0 | 12.0 | 12.0 |
| Wheat flour | 16.5 | 15.8 | 15.2 | 14.6 | 13.9 | 13.3 |
| Fish oil [3] | 2.0 | 2.0 | 2.0 | 2.0 | 2.0 | 2.0 |
| Soybean oil | 2.0 | 1.8 | 1.6 | 1.4 | 1.2 | 1.0 |
| Vitamin mix [4] | 1.0 | 1.0 | 1.0 | 1.0 | 1.0 | 1.0 |
| Mineral mix [5] | 1.0 | 1.0 | 1.0 | 1.0 | 1.0 | 1.0 |
| Choline | 0.5 | 0.5 | 0.5 | 0.5 | 0.5 | 0.5 |
| Nutrients (%, DM) | | | | | | |
| Dry matter | 94.3 | 94.1 | 94.0 | 94.3 | 94.2 | 94.3 |
| Crude protein | 55.3 | 55.3 | 55.6 | 55.1 | 55.5 | 55.3 |
| Crude lipid | 10.3 | 10.5 | 10.9 | 10.4 | 10.8 | 10.7 |
| Ash | 12.4 | 12.5 | 13.0 | 12.9 | 13.4 | 12.5 |

[1] Fish meal (FM) (crude protein: 72.1%, crude lipid: 9.1%, ash: 16.0%) was a blend of anchovy meal and sardine meal at a ratio of 1:1 (USD 1.35/kg FM, USD 1 = KRW 1336 (Korean currency)). [2] Meat meal (MM) (crude protein: 65.0%, crude lipid: 10.9%, ash: 17.9%) was purchased from Daekyung Oil & Transportation Co., Ltd. (Busan Metropolitan City, Korea) (USD 0.89/kg MM). [3] Fish oil was purchased from Ewha Oil and Fat Industry Co. Ltd. (Busan Metropolitan City, Korea). [4] Vitamin mix contained the following amounts, which were diluted in cellulose (g/kg mix): L-ascorbic acid, 200; α-tocopheryl acetate, 20; thiamine hydrochloride, 5; riboflavin, 8; pyridoxine, 2; niacin, 40; Ca-D-pantothenate, 12; myo-inositol, 200; D-biotin, 0.4; folic acid, 1.5; p-amino benzoic acid, 20; $K_3$, 4; A, 1.5; $D_3$, 0.003; cyanocobalamin, 0.003. [5] Mineral mix contained the following ingredients (g/kg mix): NaCl, 7; $MgSO_4 \cdot 7H_2O$, 105; $NaH_2PO_4 \cdot 2H_2O$, 175; $KH_2PO_4$, 224; $CaH_4(PO_4)_2 \cdot H_2O$, 140; ferric citrate, 17.5; $ZnSO_4 \cdot 7H_2O$, 2.8; Ca-lactate, 21.8; CuCl, 0.2; $AlCl_3 \cdot 6H_2O$, 0.11; $KIO_3$, 0.05; $Na_2Se_2O_3$, 0.007; $MnSO_4 \cdot H_2O$, 1.4; $CoCl_2 \cdot 6H_2O$, 0.07.

### 2.2. Experimental Fish and Culturing Conditions

Juvenile olive flounder were acquired from the Seoul hatchery (Taean-gun, Chungcheongnam-do, Korea). Prior to the feeding experiment, fish were acclimated to the culturing conditions for 14 days. The fish were fed with a commercial extruded pellet containing 55.0% crude protein and 8.0% crude lipid (National Federation of Fisheries Cooperative Feed, Gyeongsangnam-do, Korea) during the acclimatization period. After the 14-day acclimatization period, 360 juvenile fish (initial weight of 14.7 g) were randomly allocated into 18 50-L flow-through containers (water volume: 40 L). Each container was stocked with 20 juvenile fish, and all experimental diets were fed to triplicate groups of fish. Sand-filtered seawater was supplied to each container throughout the whole feeding trial, and the flow rate of seawater into each container was 4.4 L/min. The aeration provided to each container by the sand aerator and the photoperiod was in accordance with natural conditions. The water quality was monitored daily using a multiple water quality meter (AZ-8603, AZ Instrument, Taiwan). During the feeding trial, the water temperature varied from 17.8 to 27.1 °C (mean ± SD: 22.0 ± 1.61 °C), the dissolved oxygen varied from 9.2 to 10.2 mg/L, the salinity ranged from 32.0 to 32.5 g/L, and the pH varied from 7.9 to 8.1. The fish were hand-fed to satiation twice a day (08:00 and 17:00), seven days a week for 56 days. All containers were cleaned, and dead fish were removed immediately when observed.

### 2.3. Sample Collection and Biological Measurements of the Experimental Fish

Following the completion of the feeding trial, all surviving fish were fasted for 24 h, and then anesthetized with tricaine methanesulfonate (MS222) at a concentration of 100 mg/L. All fish from each container were counted and weighed collectively. Eight anesthetized fish were randomly chosen from each container to measure their condition factor (CF),

viscerosomatic index (VSI), and hepatosomatic index (HSI). The growth metrics of the fish were calculated using the following equations: specific growth rate (SGR, %/day) = [(Ln final weight of fish − Ln initial weight of fish) × 100]/days of feeding (56 days); feed efficiency ratio (FER) = (total final weight of fish + total weight of dead fish − total initial weight of fish)/total feed consumption; protein efficiency ratio (PER) = weight gain of fish/protein fed; protein retention (PR, %) = protein gain of fish × 100/protein fed; CF (g/cm$^3$) = body weight (g) × 100/total length (cm)$^3$; VSI (%) = visceral weight × 100/body weight; HSI (%) = liver weight × 100/body weight.

### 2.4. Hematological Analysis of Olive Flounder

Three anesthetized fish were chosen at random from each container, and blood samples were collected from the caudal veins with a heparinized syringe. The blood samples were centrifuged for 10 min at 2700× *g*, and the obtained plasma was stored at −70 °C for analysis of the aspartate aminotransferase (AST), alanine aminotransferase (ALT), alkaline phosphatase (ALP), total bilirubin (T-BIL), total cholesterol (T-CHO), triglyceride (TG), total protein (TP), and albumin (ALB) using an automatic chemistry system (Fuji Dri-Chem NX500i, Fujifilm, Tokyo, Japan).

### 2.5. Innate Immune Responses of Olive Flounder

Three anesthetized fish were chosen at random from each container, and blood samples were collected from the caudal veins. Serum samples were separated by centrifugation at 2700× *g* for 10 min and stored at −70 °C. Following the manufacturer's recommendations, the percentage reaction inhibition rate of the enzyme with water-soluble tetrazolium dye (WST-1) substrate and xanthine oxidase was used to measure the activity of superoxide dismutase (SOD). Following a 20-min reaction period at 37 °C, absorbance at 450 nm (the absorbance wavelength for the colored product of the WST-1 reaction with superoxide) was used to monitor each endpoint. The percentage inhibition was normalized by mg of protein and presented as SOD activity units.

A lysozyme turbidimetric assay was carried out in accordance with the study of [23]. In brief, 100 µL of the test serum was mixed with 1.9 mL of a suspension of *Micrococcus lysodeikticus* (0.2 mg/mL; Sigma, St. Louis, MO, USA) in a 0.05 M sodium phosphate buffer at pH 6.2. The reaction was conducted at 25 °C, and the absorbance at 530 nm was measured using a spectrophotometer after 0 and 60 min. A unit of lysozyme activity was defined as the amount of enzyme required to produce a 0.001/min reduction in absorbance.

### 2.6. Determination of the Chemical Composition of Olive Flounder and Experimental Diets

The rest of the surviving fish (≥3) from each container and the experimental diets were homogenized and used for chemical analysis. The Association of Official Analytical Chemists (AOAC)'s standard procedures [24] were used to measure the chemical composition of the experimental diets and whole-body fish. Acid digestion with the Kjeldahl method (Kjeltec 2100 distillation unit, Foss Tecator, Hoganas, Sweden) was used to analyze the crude protein content (N × 6.25). The crude lipid content was measured by the ether extraction method using the Soxtec TM 2043 Fat Extraction System (Foss Tecator, Hoganas, Sweden). The moisture was estimated using an oven-drying process at 105 °C for 24 h. The ash content was measured after 4 h of combustion in a muffle furnace at 550 °C.

### 2.7. Statistical Analysis

The differences among the means of the treatments were tested by one-way ANOVA and Duncan's multiple range test [25] using the SPSS program (version 26.0, Chicago, IL, USA). Before statistical analysis, all percentage data were transformed into arcsine. Furthermore, regression analysis was performed between the weight gain, SGR, FER, and PER of the fish as the dependent variables and the replaced levels of MM for FM in the diets as the independent variable.

## 3. Results

### 3.1. Performance of Fish in the Feeding Trial

The survival ($\geq$95%) of the fish was not noticeably impacted ($p > 0.3$) by the dietary replacement levels of FM with MM (Table 2). However, the weight gain and SGR of the fish fed the MM0 and MM10 diets were statistically greater ($p < 0.001$ for both) than those of the fish fed the MM30, MM40, and MM50 diets, but not statistically different ($p > 0.05$) from those of the fish fed the MM20 diet. Regarding the orthogonal polynomial contrast, the weight gain and SGR of the fish showed significant linear relationships ($p < 0.0001$ for both) with the dietary replacement of MM for FM. The best fitting models between the dietary substitution levels of MM for FM and weight gain ($Y = 0.00009395X^3 - 0.007644X^2 + 0.09283X + 40.6283$, $R^2 = 0.7865$, $p < 0.001$) and SGR ($Y = 0.000003159X^3 - 0.0002562X^2 + 0.003123X + 2.3647$, $R^2 = 0.7755$, $p < 0.001$) were observed (Table 3). To incite the maximum values of weight gain and the SGR of the fish, an estimated 7.0% of FM substitution with MM was required in the diets.

**Table 2.** Survival (%), weight gain (g/fish), and specific growth rate (SGR) of olive flounder fed experimental diets for 56 days.

| Experimental Diets | Initial Weight (g/fish) | Final Weight (g/fish) | Survival (%) | Weight Gain (g/fish) | SGR [1] (%/day) |
|---|---|---|---|---|---|
| MM0 | 14.7 ± 0.00 | 55.3 ± 0.35 | 97.5 ± 1.14 | 40.6 ± 0.35 [a] | 2.36 ± 0.011 [a] |
| MM10 | 14.7 ± 0.00 | 55.6 ± 0.49 | 96.7 ± 1.67 | 40.9 ± 0.50 [a] | 2.37 ± 0.016 [a] |
| MM20 | 14.7 ± 0.00 | 54.7 ± 0.40 | 100.0 ± 0.00 | 40.0 ± 0.40 [ab] | 2.34 ± 0.013 [ab] |
| MM30 | 14.7 ± 0.01 | 54.0 ± 0.49 | 95.0 ± 2.89 | 39.3 ± 0.49 [b] | 2.32 ± 0.016 [bc] |
| MM40 | 14.7 ± 0.01 | 52.7 ± 0.37 | 98.3 ± 1.67 | 38.0 ± 0.37 [c] | 2.28 ± 0.013 [c] |
| MM50 | 14.7 ± 0.01 | 52.7 ± 0.31 | 100.0 ± 0.00 | 38.0 ± 0.31 [c] | 2.28 ± 0.010 [c] |
| *p*-value | | | >0.3 | <0.001 | <0.001 |

Values (means of triplicate ± SE) in the same column with different letters are significantly different ($p < 0.05$).
[1] SGR (%/day) = [(ln final weight of fish − ln initial weight of fish) × 100]/days of feeding trial (56 days).

**Table 3.** Regression analysis of dietary substitution level of fish meal with meat meal as independent variable vs. parameters (weight gain, specific growth rate (SGR), feed efficiency ratio (FER), and protein efficiency ratio (PER) of olive flounder) as dependent variables.

| Dependent Variables | Orthogonal Polynomial Contrast [a] | | | Regression Analysis | | | |
| | Linear | Quadratic | Cubic | Equation | *p*-Value | R² | Y$_{max}$ (%) |
|---|---|---|---|---|---|---|---|
| Weight gain | 0.0001 | 0.3824 | 0.0894 | $Y = 0.00009395X^3 - 0.007644X^2 + 0.09283X + 40.6283$ | <0.001 | 0.7865 | X = 7.0 |
| SGR | 0.0001 | 0.4056 | 0.0929 | $Y = 0.000003159X^3 - 0.0002562X^2 + 0.003123X + 2.3647$ | <0.001 | 0.7755 | X = 7.0 |
| FER | 0.0001 | 0.8686 | 0.1050 | $Y = 0.000002158X^3 - 0.0001594X^2 + 0.001406X + 1.0158$ | <0.001 | 0.7540 | X = 4.9 |
| PER | 0.0034 | 0.2494 | 0.0457 | $Y = 0.000006433X^3 - 0.0004363X^2 + 0.004538X + 1.7929$ | <0.009 | 0.5290 | X = 6.0 |

[a] If statistical significance ($p < 0.05$) was detected, the model that fit best with the data was selected Y$_{max}$ (%) indicates the meat meal substitution level for fish meal in diets to achieve the maximum for each dependent variable.

The feed consumption of the fish was not statistically influenced ($p > 0.1$) by the dietary replacement of FM with MM (Table 4). The FERs of the fish fed the MM0 and MM10 diets were statistically higher ($p < 0.002$) than those of the fish fed the MM30, MM40, and MM50 diets, but not noticeably different ($p > 0.05$) from that of the fish fed the MM20 diet. The PERs of the fish fed the MM0, MM10, and MM20 diets were statistically higher ($p < 0.02$) than those of the fish fed the MM30 and MM40 diets, but not noticeably different ($p > 0.05$) from that of the fish fed the MM50 diet. In terms of the orthogonal polynomial contrast, the FERs and PERs of the fish demonstrated statistical linear relationships ($p < 0.0001$ and $p < 0.0034$, respectively) with the dietary replacement of MM for FM. The best-fitting models between the dietary substitution levels of MM for FM and the FER ($Y = 0.000002158X^3 - $

$0.0001594X^2 + 0.001406X + 1.0158$, $R^2 = 0.7540$, $p < 0.001$) and the PER ($Y = 0.000006433X^3 - 0.0004363X^2 + 0.004538X + 1.7929$, $R^2 = 0.5290$, $p < 0.009$) were observed. To achieve the maximum values of the FER and PER of the fish, the estimated rates of 4.9% and 6.0%, respectively, of FM substitution with MM were required in the diets. The PR of the fish was not noticeably influenced ($p > 0.05$) by the dietary replacement of FM with MM. The CF of the fish fed the MM0 diet was statistically higher ($p < 0.04$) than that of the fish fed the MM20 and MM40 diets, but not noticeably different ($p > 0.05$) from that of the fish fed the MM10, MM30, and MM50 diets. However, neither the VSI nor the HSI of the fish was statistically altered ($p > 0.9$ and $p > 0.4$, respectively) by the dietary replacement of FM with MM.

**Table 4.** Feed consumption (g/fish), feed efficiency ratio (FER), protein efficiency ratio (PER), protein retention (PR), condition factor (CF), viscerosomatic index (VSI), and hepatosomatic index (HSI) of olive flounder fed the experimental diets for 56 days.

| Experimental Diets | Feed Consumption (g/fish) | FER [1] | PER [2] | PR [3] | CF [4] | VSI [5] | HIS [6] |
|---|---|---|---|---|---|---|---|
| MM0 | 41.0 ± 0.62 | 1.02 ± 0.008 [a] | 1.79 ± 0.018 [a] | 33.90 ± 0.938 | 1.29 ± 0.002 [a] | 3.16 ± 0.002 | 1.22 ± 0.001 |
| MM10 | 41.4 ± 0.70 | 1.02 ± 0.011 [a] | 1.79 ± 0.013 [a] | 31.49 ± 1.397 | 1.29 ± 0.001 [ab] | 3.16 ± 0.000 | 1.22 ± 0.001 |
| MM20 | 40.2 ± 0.08 | 1.00 ± 0.010 [ab] | 1.79 ± 0.018 [a] | 31.72 ± 1.915 | 1.28 ± 0.000 [b] | 3.16 ± 0.000 | 1.22 ± 0.000 |
| MM30 | 42.3 ± 1.22 | 0.97 ± 0.011 [bc] | 1.68 ± 0.035 [b] | 28.81 ± 2.669 | 1.28 ± 0.000 [ab] | 3.16 ± 0.001 | 1.22 ± 0.000 |
| MM40 | 40.3 ± 0.68 | 0.96 ± 0.009 [c] | 1.70 ± 0.031 [b] | 29.38 ± 2.291 | 1.28 ± 0.000 [b] | 3.16 ± 0.000 | 1.22 ± 0.000 |
| MM50 | 39.6 ± 0.04 | 0.96 ± 0.009 [c] | 1.73 ± 0.016 [ab] | 29.55 ± 2.187 | 1.28 ± 0.001 [ab] | 3.16 ± 0.000 | 1.22 ± 0.000 |
| *p*-value | >0.1 | < 0.002 | < 0.02 | >0.3 | <0.04 | >0.9 | >0.4 |

Values (means of triplicate ± SE) in the same column with different letters are significantly different ($p < 0.05$). [1] Feed efficiency ratio (FER) = (total final weight of fish + total weight of dead fish − total initial weight of fish)/total feed consumption. [2] Protein efficiency ratio (PER) = weight gain of fish/protein fed. [3] Protein retention (PR) = protein gain × 100/protein fed. [4] Condition factor (CF, g/cm$^3$) = body weight (g) × 100/total length (cm)$^3$. [5] Viscerosomatic index (VSI, %) = visceral weight × 100/body weight. [6] Hepatosomatic index (HSI, %) = liver weight × 100/body weight.

### 3.2. Proximate Composition of the Whole-Body Olive Flounder

The moisture contents of the whole bodies of the fish fed the MM50 diet were statistically higher ($p < 0.04$) than those of the fish fed the MM0 and MM30 diets, but not noticeably different ($p > 0.05$) from those of the fish fed the MM10, MM20, and MM40 diets (Table 5). The crude protein content of the whole-body fish varied from 15.8 to 16.5% and the ash content varied from 3.4 to 3.7%, but these parameters were not noticeably altered ($p > 0.9$ and $p > 0.3$, respectively) by the dietary replacement of FM with MM. The fish fed the MM40 and MM50 diets showed statistically higher ($p < 0.002$) crude lipid contents than the fish fed all other diets.

**Table 5.** Proximate composition (%) of the whole body of olive flounder fed experimental diets for 56 days.

| Experimental Diets | Moisture | Crude Protein | Crude Lipid | Ash |
|---|---|---|---|---|
| MM0 | 74.7 ± 0.11 [b] | 16.5 ± 0.09 | 3.0 ± 0.01 [b] | 3.4 ± 0.01 |
| MM10 | 75.6 ± 0.84 [ab] | 16.3 ± 0.66 | 3.0 ± 0.05 [b] | 3.4 ± 0.15 |
| MM20 | 76.6 ± 0.62 [ab] | 16.3 ± 0.68 | 3.1 ± 0.05 [b] | 3.6 ± 0.08 |
| MM30 | 75.0 ± 0.16 [b] | 16.4 ± 0.18 | 3.0 ± 0.02 [b] | 3.5 ± 0.09 |
| MM40 | 76.5 ± 0.49 [ab] | 15.9 ± 0.76 | 3.2 ± 0.03 [a] | 3.4 ± 0.13 |
| MM50 | 77.2 ± 0.88 [a] | 15.8 ± 0.81 | 3.2 ± 0.02 [a] | 3.7 ± 0.14 |
| *p*-value | <0.04 | >0.9 | <0.002 | >0.3 |

Values (means of triplicate ± SE) in the same column with different letters are significantly different ($p < 0.05$).

### 3.3. Plasma Chemistry of Olive Flounder

The plasma AST varied from 18.1 to 18.7 U/L, the ALT varied from 5.8 to 5.9 U/L, the ALP varied from 156.8 to 157.9 U/L, the T-CHO varied from 239.1 to 241.0 mg/dL, the

TG varied from 386.6 to 389.4 mg/dL, and the ALB varied from 0.8 to 1.0 g/dL (Table 6). In addition, the T-BIL and TP rates were 0.3 mg/dL and 3.9 g/dL, respectively. None of the plasma parameters of the fish were noticeably impacted ($p > 0.05$) by the experimental diets.

**Table 6.** Hematological parameters of olive flounder fed the experimental diets for 56 days.

| Experimental Diets | AST (U/L) | ALT (U/L) | ALP (U/L) | T-BIL (mg/dL) | T-CHO (mg/dL) | TG (mg/dL) | TP (g/dL) | ALB (g/dL) |
|---|---|---|---|---|---|---|---|---|
| MM0 | 18.1 ± 0.61 | 5.9 ± 0.34 | 156.8 ± 8.38 | 0.3 ± 0.03 | 239.1 ± 3.93 | 389.1 ± 4.24 | 3.9 ± 0.05 | 1.0 ± 0.03 |
| MM10 | 18.2 ± 0.70 | 5.8 ± 0.22 | 157.9 ± 4.01 | 0.3 ± 0.03 | 241.0 ± 5.18 | 388.3 ± 5.77 | 3.9 ± 0.06 | 0.9 ± 0.04 |
| MM20 | 18.3 ± 0.50 | 5.9 ± 0.20 | 156.8 ± 4.57 | 0.3 ± 0.02 | 240.7 ± 5.42 | 389.4 ± 7.37 | 3.9 ± 0.06 | 1.0 ± 0.04 |
| MM30 | 18.7 ± 0.73 | 5.9 ± 0.31 | 156.9 ± 6.17 | 0.3 ± 0.03 | 240.3 ± 4.73 | 387.7 ± 6.29 | 3.9 ± 0.05 | 0.9 ± 0.03 |
| MM40 | 18.3 ± 0.93 | 5.8 ± 0.22 | 156.8 ± 4.68 | 0.3 ± 0.03 | 240.7 ± 5.25 | 387.7 ± 5.52 | 3.9 ± 0.04 | 0.9 ± 0.05 |
| MM50 | 18.3 ± 0.65 | 5.8 ± 0.22 | 157.4 ± 4.84 | 0.3 ± 0.02 | 240.0 ± 5.14 | 386.6 ± 6.58 | 3.9 ± 0.05 | 0.8 ± 0.06 |
| *p*-value | >0.9 | >0.9 | >0.9 | >0.1 | >0.9 | >0.9 | >0.9 | >0.3 |

Values (means of triplicate ± SE) in the same column with different letters are significantly different ($p < 0.05$). AST, analysis of aspartate aminotransferase; ALT, alanine aminotransferase; ALP, alkaline phosphatase; T-BIL, total bilirubin; T-CHO, total cholesterol; TG, triglyceride; TP, total protein; and ALB, albumin.

### 3.4. Innate Immune Responses of Olive Flounder

None of the SOD or lysozyme activities of olive flounder were noticeably altered ($p > 0.6$ and $p > 0.07$, respectively) by the dietary replacement of FM with MM (Table 7).

**Table 7.** Lysozyme activity and SOD of olive flounder fed the experimental diets for 56 days.

| Experimental Diets | SOD Activity (%) | Lysozyme Activity (U/mL) |
|---|---|---|
| MM0 | 70.01 ± 1.648 | 0.144 ± 0.016 |
| MM10 | 68.06 ± 0.549 | 0.121 ± 0.024 |
| MM20 | 68.85 ± 1.256 | 0.092 ± 0.005 |
| MM30 | 67.96 ± 0.632 | 0.089 ± 0.012 |
| MM40 | 68.60 ± 0.541 | 0.094 ± 0.004 |
| MM50 | 67.96 ± 0.353 | 0.085 ± 0.015 |
| *p*-value | >0.6 | >0.07 |

Values (means of triplicate ± SE) in the same column with different letters are significantly different ($p < 0.05$). SOD, superoxide dismutase.

## 4. Discussion

MM is an economically cheaper source of protein than FM and widely accessible in meat processing plants and animal slaughterhouses. In the last few decades, huge amounts of by-products, such as skin, viscera, heads, feet, and blood have become available as a result of the modern methods of animal meat processing [26]. Recycling wastes from meat processing plants and animal slaughterhouses has economical, biological, and environmental benefits. The nutritional contents of MM, being rich in protein and lipids, makes it suitable as an ingredient in feeds for carnivorous fish [27]. This may imply that MM has the nutritional potential as a substitute for FM in olive flounder diet because it requires relatively high protein and lipid content in diets [28,29]. In fact, the suitability of terrestrial animal by-product meal as a replacement for FM in the diets of various fish species has been evaluated over the last few decades [15,17,30–32]. In addition, MM as a potential replacement for FM has been known in a few fish species, such as olive flounder, barramundi (*Lates calcarifer*), rockfish (*Sebastes schlegeli*), and Australian silver perch (*Bidyanus bidyanus*) [18,19,33,34].

The lack of differences in the growth performance (weight gain and SGR), feed consumption, and feed utilization (FER, PER, and PR) of olive flounder fed the MM0, MM10, and MM20 diets in the current study proved that MM could be successfully replaced for FM up to 20% in a 65% FM-based diet of olive flounder without deterioration in the growth, feed consumption, or feed utilization. According to the third-order regression analysis, however, the ideal replacement levels of MM for FM in diets for the utmost growth (weight

gain and SGR) of olive flounder were estimated to be 7.0% for both. Therefore, considering the economic views of the present study, replacing 20% FM with MM in the diet seems to be the most recommendable for olive flounder farmers in terms of feed cost.

The FER of olive flounder deteriorated when more than a 30% FM replacement with MM in the diet was made. In addition, the PERs of the fish fed the MM30 and MM40 diets were also inferior to the fish fed the MM0 diet in the current study. The consistent results show that the FERs and PERs of olive flounder were impacted by the dietary replacement of FM with MM [15,18]. Similarly, the feed utilization of gibel carp and Japanese sea bass (*Lateolabrax japonicus*) was influenced by the dietary replacement of FM with various animal protein sources [35,36]. The dietary replacement of FM with MM influenced both the feed consumption and feed utilization (FER, PER, and PR) of olive flounder, and the dietary replacement of 20% FM with MM achieved the best growth performance, highest feed consumption, and highest FER [16].

The maximum replacement level (20%) of FM with MM in the diet of olive flounder in the current study was relatively low compared to substitution levels reported in the same fish species in other studies. Reference [18] demonstrated that substitution of FM up to 60% with pet-grade MM with the supplementation of AAs had no negative impact on the growth, hematological parameters, or chemical composition of juvenile olive flounder (initial weight of about 3 g) when 80% FM was included. Recently, [16] unveiled that up to 40% of FM could be successfully replaced with pet-grade MM without the supplementation of AAs, producing no adverse effects on the growth performance, feed utilization, or non-specific immune responses of fish when juvenile olive flounder (initial weight of 9.2 g) were fed with a 65% FM-based diet or one of the diets replacing 10%, 20%, 40%, 60%, 80%, or 100% FM with MM for 56 days. During the winter season, up to 40% of FM could be successfully replaced with feed ingredient-grade MM without causing a reduction in the growth performance of olive flounder in a 10-week feeding trial [15]. Considering these results and the results obtained from the present study, the substitutability of FM with MM in olive flounder feed was highly influenced by the quality (or grade) of MM, the supplementation of EAAs, and other experimental conditions, such as the fish size, water temperature, and duration of the feeding trial. Furthermore, [37] indicated that porcine MM can substitute up to 35% of FM in a 42.4% FM-based diet without causing any adverse impact on the growth, chemical composition, FER, or PER of Pacific white shrimp (*Litopenaeus vannamei*). However, further study on the substitutability of FM with MM in olive flounder extruded pellets should be tested in commercial scale farms before practical implementation. The VSI and HSI of fish were not altered by the dietary substitution of MM for FM at the end of the 56-day feeding trial. Likewise, references [17,38] have also reported that the HSI and VSI were not noticeably impacted by the substitution of FM with chicken by-product meal and meat and bone meal in fish feeds.

The moisture and crude lipid contents of the whole bodies of olive flounder were influenced by the dietary replacement levels of FM with MM in the current study. The fish fed the MM50 diet showed the highest moisture and lipid contents in whole-body fish. Similarly, references [18,39] found that the chemical composition of olive flounder was altered by the dietary replacement of FM with MM and dehulled soybean meal, respectively. On the contrary, there were also some conflicting studies showing that the proximate composition of olive flounder was not impacted by the replacements of FM with various alternative protein sources in diets [10,16,38,40].

Hematological parameters are used to indicate the health conditions, physiological and nutritional status, and pathological changes in fish [41]. In the current study, however, there were no considerable differences in any of the hematological parameters measured, demonstrating that the dietary replacement of MM for FM might not alter the hematological status of olive flounder. Likewise, the dietary replacement of MM for FM did not change the hematological parameters of olive flounder [15,16]. In addition, reference [33] also demonstrated that the dietary replacement of FM with MM did not cause any difference in the total protein or glucose of rockfish.

The innate immune system is essential for disease resistance in fish. Compared to mammals, fish rely more heavily on innate defense mechanisms [42]. SOD and lysozyme activities are well-known markers of stress response and disease resistance [43]. SOD is a free radical-neutralizing enzyme that can protect animal tissue from damage [44]. Lysozyme activity is a crucial humoral indicator of the immune system that acts as a main defensive component against invasive microorganisms in fish [45]. It can catalyze the hydrolysis of β (1–4) glycosidic bonds of bacterial cell walls and plays an essential role in the defense against bacterial infections [46]. However, neither the SOD nor lysozyme activity of olive flounder was altered by the partial dietary replacement of FM with MM in the current study, demonstrating that the replacement of FM with MM in olive flounder feed had no adverse influence on their innate immune responses. Similarly, the dietary replacement of FM with various animal and plant protein sources did not bring about any adverse impact on the immune responses of olive flounder or red sea bream [9,11,16]. Unlike this study, however, the serum SOD and lysozyme activities of tilapia (*Oreochromis niloticus* × *O. aureus*) tended to decrease with increased dietary replacement levels of FM with soybean meal because the immune system of tilapia might be influenced by the certain constituents of soybean meal [47].

**5. Conclusions**

Up to 20% of FM could be successfully replaced with feed ingredient-grade MM in feed without causing adverse impacts on the growth, feed consumption, feed utilization, or innate immune responses of olive flounder. The optimum replacement levels of FM with MM for the maximum weight gain and SGR of olive flounder were estimated to be 7.0% for both.

**Author Contributions:** M.F.U.Z.; Data Curation, Formal Analysis, Investigation, Visualization. R.L.; Formal Analysis, Investigation, Visualization, Writing—Original Draft. S.H.C.; Conceptualization, Methodology, Writing—Review and Editing, Funding Acquisition. All authors have read and agreed to the published version of the manuscript.

**Funding:** This work was supported by the National Research Foundation of Korea (NRF) grant funded by the Korean Government (MSIT) (No. 2020R1A2C1009903).

**Institutional Review Board Statement:** Not applicable.

**Data Availability Statement:** The data are available upon request from the authors.

**Conflicts of Interest:** The authors declare no conflict of interest.

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
