# Peer review of "Evaluation of Meat Meal as a Replacer for Fish Meal in Diet on Growth Performance, Feed Utilization, Chemical Composition, Hematology, and Innate Immune Responses of Olive Flounder (Paralichthys olivaceus)"

_fishes, doi:10.3390/fishes7060343_

Round 1

Reviewer 1 Report

Although the simple idea of the study, the manuscript is well-designed and written clearly. Please revise the manuscript as follows:

Keywords should begin with capitals.

Line 20 & 136: what is the full definition of SGR?

It is unclear whether you feed the shrimps with sinking or floating feed.

The amino acid profile should be included in Table 1.

Lin 136: correct to "Ln".

Line 156: Please define SOD for the first use.

Line 168: define AOAC for the first use.

In Tables 6 & 7: all abbreviations should be defined fully in the table's footnote. GOT and GPT are the old names for ALT and AST; please review the whole text.

Author Response

The manuscript has been revised or corrected according to comments of the reviewer 1

Although the simple idea of the study, the manuscript is well-designed and written clearly. Please revise the manuscript as follows:

Keywords should begin with capitals.

→ No, keywords begin with lower case instead of capitals in FISHES.

Line 20 & 136: what is the full definition of SGR?

→ Both were revised.

It is unclear whether you feed the shrimps with sinking or floating feed.

→ Since the experimental diet were pressure-pelleted with a laboratory pellet extruder, all experimental diets were sinking dry pellets.

The amino acid profile should be included in Table 1.

→We are sorry, unfortunately, we do not have amino acid profiles of the experimental diets. This is another reason for us not to discuss or describe about AA a lot in the manuscript.

Lin 136: correct to "Ln".

→It was corrected.

Line 156: Please define SOD for the first use.

→It was revised.

Line 168: define AOAC for the first use.

→It was revised.

In Tables 6 & 7: all abbreviations should be defined fully in the table's footnote. GOT and GPT are the old names for ALT and AST; please review the whole text.

 →They were all revised throughout the manuscript as you suggested.

I really appreciate for your valuable comments in this manuscript.

From corresponding author

Reviewer 2 Report

An interesting paper to replace fishmeal with alternative meals,

They could attend editorial revisions on line 38 (world). Indicate in the methodology, the anticoagulant solution used in the blood extraction. Homogenize the format of the references in the bibliography section.

Author Response

The manuscript has been revised or corrected according to comments of the reviewer 2

An interesting paper to replace fishmeal with alternative meals,

They could attend editorial revisions on line 38 (world).

→ It has been corrected.

Indicate in the methodology, the anticoagulant solution used in the blood extraction.

→ It was corrected.

Homogenize the format of the references in the bibliography section.

→ All were corrected.

I really appreciate for your valuable comments in this manuscript.

From corresponding author

Round 2

Reviewer 1 Report

The paper is greatly improved and can be accepted in the current form.